# Achieving Influenza Vaccine Uptake Target in Canada via a Pharmacy-Led Telephone Discussion during the 2019–2020 Season

**DOI:** 10.3390/vaccines9040312

**Published:** 2021-03-26

**Authors:** William David Strain, James Mansi, Constantina Boikos, Michael Boivin, William A. Fisher

**Affiliations:** 1Diabetes and Vascular Research Centre, University of Exeter Medical School, Exeter EX2 5AX, UK; 2Academic Department of Healthcare for Older People, Royal Devon & Exeter Hospital, Exeter EX2 5DW, UK; 3Centre for Outcomes Research & Evaluation (CORE), Seqirus, QC H9H 4M7, Canada; James.Mansi@Seqirus.com (J.M.); Constantina.Boikos@Seqirus.com (C.B.); 4Independent Pharmacist Consultant, Barrie, ON L4N 6Z6, Canada; mike@commpharm.com; 5Department of Obstetrics and Gynaecology, Western University Canada, London, ON N6A 5C2, Canada; fisher@uwo.ca

**Keywords:** influenza, vaccine, immunisation, vaccine hesitancy

## Abstract

Older adults (≥65 years) are at elevated risk of influenza-related morbidity and mortality. Many developed countries do not achieve the World Health Organization influenza immunization target of 75% in people ≥65 years. We aimed to determine whether a brief pharmacy phone call could increase vaccine uptake of standard and enhanced influenza. Twenty-eight community pharmacists across Canada performed a telephone consultation with 643 older adults whose primary care records indicated that they had not received their influenza vaccination from their usual practitioner. Of these 643 adults, 169 (26.3%) had been vaccinated in another setting. Of the remaining 474, 313 (66%) agreed to receive the vaccine. Of those who refused vaccination, 69 provided a rationale for not wanting it, including that the flu shot “causes the flu” (*n* = 25), “doesn’t work” (*n* = 25), “is too painful” (*n* = 10), and other (*n* = 10). Overall, of the 643 individuals who had not received their vaccination from their usual health care provider in the first wave of vaccinations, 75.4% (*n* = 485) ultimately received their vaccination in the 2019–2020 season. This highlights the important role of the community pharmacist in achieving the World Health Organization (WHO) targets for vaccination.

## 1. Introduction

The National Advisory on Immunization (NACI) estimates that there are an average of 12,200 hospitalizations related to influenza and approximately 3,500 deaths attributable to influenza annually in Canada [1]. Public Health England estimates that, on average, 17,000 people have died of influenza in England alone annually since 2014. In the USA, the Center for Disease Control and Prevention (CDC) has estimated the numbers of influenza cases ranged from 9–45 million a year and deaths from 12,000 to 61,000 in the same time frame. The majority (~90%) of influenza-related deaths in the 2018–2019 flu season in the US were in older adults (>65 years) and, among older survivors, it was determined that 13% will experience catastrophic disability and face the prospect of loss of independence and quality of life [2]. As such, individuals ≥65 years of age form the optimal age group to target for preventive measures, with demonstrable personal and health economic benefits by achieving reductions in cases in this vulnerable population.

Annual vaccination is arguably the most effective public health strategy for preventing influenza. National immunization programs across the world prioritise achieving optimal influenza vaccination coverage in older and vulnerable adults [1,3,4]. In this connection, the World Health Organization (WHO) recommends that 75% of those 65 years of age and older receive an influenza vaccine [5]. Many countries set slightly different targets for this age group, such as Canada, who has an 80% influenza immunization target for this age group [6]. In recent years, however, the United States, Canada and the United Kingdom have not achieved these targets [7,8,9,10,11,12].

The reasons for failure to attain benchmark influenza vaccine coverage are complex and multifactorial. Proposed explanations include a perceived lack of salience, lack of access to vaccines, a belief in common influenza vaccine myths such as the belief that influenza is not a serious health concern, or a belief that the influenza vaccine is ineffective and/or causes the disease [13,14,15]. Furthermore, there remain unprecedented additional barriers to immunization that are posed due to the pandemic co-circulation of SARS-CoV-2 and influenza in the upcoming 2020–2021 influenza season in the Northern Hemisphere. Physical distancing and shielding of high-risk individuals may reduce the number of individuals that present for routine vaccinations including seasonal influenza. New ways of working in primary and secondary care, including widespread reliance on telemedicine, may reduce capacity for in-person vaccination clinics and may leave large numbers of people without usual access to vaccinations. With the risk of a second wave of COVID-19, the potential for a dual viral outbreak (influenza and coronavirus) could have devastating results in vulnerable populations such as those ≥65 years of age [16,17].

Over and above these obstacles to optimal influenza vaccine coverage in older adults is evidence of an approximate 50% decrease in effectiveness of standard, egg-derived vaccines in the older age group compared to younger adults [1]. This is thought to be largely due to immunosenescence, an age-related functional decline of innate and acquired immune systems that results in suboptimal humoral responses to vaccination [18]. Two enhanced vaccine options, however, have been introduced to increase immune response and improve outcomes in older adults. The high-dose trivalent influenza vaccine (TIV-HD) has been shown to reduce the number of influenza cases and hospitalization due to pneumonia in older adults compared to standard dose influenza vaccine (TIV) [19,20,21]. Furthermore, influenza vaccine with MF59 adjuvant (TIV-adj) has been shown to reduce the number of influenza cases, hospitalization due to pneumonia, and fewer cases of acute coronary and cerebrovascular event hospitalization in older adults compared to standard dose influenza vaccine (TIV) [22,23,24,25,26,27,28].

Pharmacists and their pharmacy teams are a growing resource that can support primary care in a wide variety of roles, and these professionals have the confidence of the older adult population [29,30]. Their roles in medications review, chronic disease management, administering immunizations, and providing educational engagement have demonstrated improved clinical outcomes in a range of conditions [31,32,33,34]. We hypothesised that engagement between pharmacists and individuals at high risk of severe influenza disease would help to increase influenza vaccine coverage in this population, ultimately helping to achieve the 75% benchmark vaccine coverage established by the WHO. Here, we present the outcomes of a quality improvement program aimed at increasing influenza immunization in individuals ≥65 years of age during the 2019–2020 influenza season in three Canadian provinces by facilitating conversations about influenza vaccination with consumer-facing pharmacy staff. A process evaluation of the intervention was also undertaken to determine whether a greater knowledge of available interventions would influence patient choices when considering enhanced influenza vaccine options.

## 2. Methods

Pharmacies in the Canadian provinces of Alberta, British Columbia, and Saskatchewan were approached for participation in a quality improvement program. These pharmacies were all part of one pharmacy chain with a pharmacy director who actively encouraged participation in the program.

A telephone script was created to guide a brief (target 1–3 min) and effective discussion on influenza vaccination between pharmacy staff and unvaccinated individuals ≥65 years of age. An accompanying patient education package for use in these discussions was developed in conjunction with health care practitioners, pharmacists and potential patients (Appendix A). The community pharmacists did not receive any incentive beyond the usual government reimbursement for the administration of the flu vaccines for participation in this program.

### 2.1. Participants

The participants were selected through printing a report of individuals ≥65 years who had not received an influenza vaccine dose for the 2019–2020 season from the pharmacy dispensing software. Pharmacists, pharmacy assistants, or pharmacy technicians telephoned individuals in this report. The telephone call was designed to take 1 to 3 min, and the intervention was designed to be integrated into the routine practice of the pharmacy. The primary outcome in this quality improvement programme was for those contacted to have an appointment for influenza vaccine administration arranged by the end of the telephone consultation. If a patient declined to receive the influenza vaccine, they were asked for their reason(s). The pharmacy team was provided with patient education information (Appendix A) to address common reasons for vaccine hesitancy to try and improve vaccine uptake in this age group.

As a secondary process evaluation outcome, we evaluated the impact of information contained in these scripted conversations concerning enhanced vaccine options for those ≥65 years of age. These vaccines are not reimbursed as part of the routine influenza immunization program in Canada when administered in the pharmacy and were subject to individual payment. We provided supporting information for pharmacists aimed at informing patients about enhanced vaccine options and asked participants if they would like to receive one of these enhanced vaccines.

### 2.2. Statistical Analysis

This is a secondary analysis of a quality improvement programme. Therefore, no formal statistical calculations (such as a power calculation) were performed. Rather, descriptive statistics on the cohort of individuals identified as part of the quality improvement program are presented.

### 2.3. Patient and Public Involvement

The concept of pharmacist involvement in order to improve adherence to pharmacological therapies has been evaluated with older adults with long-term diseases in several different disease areas by this group. A focus group of older adults contemplating vaccination was interviewed regarding their perception of a pharmacist-based telephone consultation. The population were welcoming of engagement with these health care professionals with regard to delivery of medical interventions, as long as they were not expected to make new clinical diagnoses.

The telephone script was developed in conjunction with a patient focus group of appropriate age to receive the intervention. The primary goal of this was to evaluate the effectiveness and acceptability of the telephone conversation in addition to the ability to comprehend of the language used.

## 3. Results

Overall, this program was implemented in 28 pharmacies in the Canadian provinces of Alberta, British Columbia, and Saskatchewan. Six hundred and forty-three individuals were identified, all of whom were at risk for severe influenza disease based on their age (≥65 years) and the lack of a pharmacy record of influenza immunization at the time the telephone call was placed. Individuals were contacted predominantly at the “tail end” of the vaccination season in order to target those who did not respond to initial invitations from their primary care physicians, from 7 October 2019 until 26 January 2020 (Figure 1). The telephone script is presented in Appendix A. Our process evaluation targeted most older adults during

December and January and thus, by definition, captures individuals who had missed or chosen not to participate in the initial annual influenza immunization clinics.

### 3.1. Evaluation of Primary Outcome

In this population of 643 individuals who did not have a documented influenza vaccine from their primary provider, 169 (26.3%) had been vaccinated in another setting. Of the remaining 474, 316 (67%) agreed to receive the vaccine by the end of the telephone consultation (Figure 2). Thus, in total, 485 (75.4%) people received their influenza vaccine by the end of this intervention. As such, this quality improvement project demonstrated its primary target of achieving the WHO recommended 75% influenza vaccine uptake in older adults who had not been vaccinated by their usual care provider in the first wave.

### 3.2. Reason for Declining the Influenza Vaccine and Efficacy of Correcting Myth

One hundred and sixty-one older adults who had not received an influenza vaccine from another healthcare professional initially declined to receive the vaccine. Of the 161 who declined the vaccine, 69 gave a reason (Table 1). The pharmacy team was able to address the reason for not receiving the vaccine for 31 of these individuals, people with a simple information script.

### 3.3. Interest in Receiving an Enhanced Influenza Vaccine

All older adults who agreed to receive the influenza vaccine were asked if they would like to receive the unadjuvanted quadrivalent vaccine that was part of the routine influenza vaccine at no cost or an enhanced vaccine that they would have purchase. By the end of the telephone conversation, 162 (52%) of participants had made a definite choice. Of these, 64 (39.5%) selected the enhanced vaccine at additional personal cost, and a further 40 individuals were amenable to receiving more details during their consultation at the pharmacy.

## 4. Discussion

Findings from this project support the utility of a simple pharmacy-led telephone engagement tool to increase vaccination uptake in older adults, thereby helping to achieve the World Health Organization’s target of 75% vaccination against influenza in this age group.

There are several factors affecting influenza vaccine uptake in older adults. Even with the benefits of influenza immunization being clearly demonstrated, some reluctant to receive the vaccine [35]. Barriers to influenza vaccination include a lack of perceived severity of influenza and concerns about the effectiveness of vaccines [35]. Some older adults may also be fearful of adverse effects, real or perceived, associated with the influenza vaccine [35]. Population reviews found that patients who are part of a visible minority group have lower household income, poor physical activity, lack of access to healthcare resources, and better overall health were more likely to not receive an influenza vaccine [36,37,38]. There is observational data demonstrating that having severe functional limitations (severe frailty) increases the risk of death by 13-fold but decreases the likelihood of influenza immunization [39]. Finally, a lack of awareness of the importance of vaccination and the benefits of self-care, or a simple failure to prioritise self-care may also be contributing factors to patients who do not see their healthcare professional during the immunization season failing to receive the vaccine.

This short pharmacy quality improvement evaluation supports the notion that the majority of older adults are receptive to receiving the influenza vaccine after a short telephone conversation. Indeed, of individuals who had not received influenza vaccination, two thirds were easily persuaded to receive this potentially lifesaving intervention. This suggests that a primary reason for not receiving the influenza vaccine is not vaccine hesitancy but rather simple inertia—failing to prioritise an individual’s own health. Vaccine hesitancy was an issue in a small number of older adults. The most common reasons were the beliefs that the influenza vaccine is not effective or that it causes influenza. Some older adults were receptive to hear a short pre-written script to address their reason for not wanting to receive the influenza vaccine; however, education regarding the myth was only effective in 3 (4.3% of those who provided a reason for declining the vaccine). The pharmacists were not trained on how to deliver the script per-se, which may also have affected the quality of the intervention in terms of addressing patients’ hesitancy. Further research in this connection is warranted.

A lack of belief in the efficacy of the vaccine was cited by several older adults as a reason for vaccine hesitancy. Interestingly, when presented with information about enhanced influenza vaccine options for their age group, approximately two fifths of individuals opted for the enhanced vaccine at additional personal cost. In some regions of the world, we note enhanced influenza vaccines are the standard of care for older adults [40]. During the 2019–2020 influenza vaccine season, enhanced vaccines administered by pharmacists to outpatients in Canada were not part of the routine influenza immunization program. Our data demonstrates that a significant portion of older adults are interested in receiving the enhanced influenza vaccine or at least learning more. We believe that older adults should be provided the option of an enhanced influenza vaccine and be allowed to decide which influenza vaccine option is suitable for their circumstance.

The influenza season in the Northern Hemisphere typically starts in November and ends in March [1]. Most influenza immunization programs, clinics, and vaccine promotion are targeted in October and November to immunise people before the onset of the influenza season. Our process evaluation targeted most older adults during December and January and thus, by definition, captures individuals who had missed or chosen not to participate in the initial annual influenza immunization clinics. This method targets ‘’salvage vaccinations’’ who were missed through other influenza public health initiatives. Therefore, we are exploring patients who, although eligible for treatment, have not engaged with the immunization program.

Pharmacist influenza immunization has been studied broadly. Pharmacist immunization has been found to increase vaccine uptake [31,41,42] and is a cost-effective consultation that leads to improved clinical outcomes [43]. Patients found that the pharmacist-based immunization was a positive experience [44], accessible with high patient favourability [30]. Our intervention contributes to this literature by exploring and demonstrating benefits in older adults that do not receive their vaccine in the “first round” and therefore are, almost by definition, vaccine resistant. The finding that the majority of these older adults are receptive to receiving influenza immunization when contacted by their pharmacy is encouraging. This is particularly important in the coming 2020–2021 “twindemic” of seasonal influenza with a second wave of COVID-19.

## 5. Limitations

There are several limitations to our process evaluation. The intervention was conducted via telephone and captures as its endpoint scheduled commitment to receive the vaccine and not observed vaccination per se. It also included a small subset of provinces and pharmacies in Canada. We were not able to capture non-responsiveness, or the number of people targeted versus who we connected with. This needs to be addressed in a prospective cluster randomised study. With regards to those receiving enhanced vaccination with either quadrivalent or adjuvant vaccination, the longer-term outcomes are not recorded. Studies with the enhanced vaccines in older adults have been shown to reduce influenza cases, hospitalizations due to pneumonia, and fewer cases of acute cases of acute coronary and cerebrovascular event hospitalizations [19,22,23,24].

Although there are several limitations, there are several strengths. This program is a simple and short intervention that can easily implemented into the average pharmacy. With a small-time commitment, pharmacists were able to ensure a large portion of older adults were immunised against influenza; added to ambient vaccination, levels this program achieved the WHO influenza vaccine coverage benchmark.

## 6. Conclusions

A short and practical pharmacy intervention in three provinces in Canada was able to reach the WHO 75% influenza immunization target for older adults. A small amount of time and cost could increase influenza immunization in this high-priority population. With the immunization issues during the COVID-19 pandemic, this program may be able to ensure that individuals are protected while COVID-19 is circulating in the community. This process evaluation demonstrated a significant impact on influenza immunization rates in older adults as well as impact of information about enhanced vaccines. We believe this proof of concept justifies further evaluations in different regions of the world and strongly supports the role of the pharmacy in influenza immunization in older adults.

## Figures and Tables

**Figure 1 vaccines-09-00312-f001:**
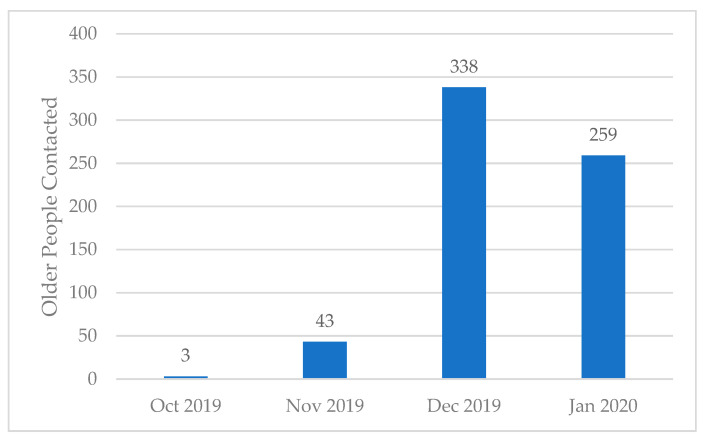
Time in the flu season when individuals were contacted. Note: the usual vaccination season commences in Mid-September in Canada. These individuals were contacted later in the flu-season in order to address vaccine hesitancy among those that had not responded to their initial primary care invitations.

**Figure 2 vaccines-09-00312-f002:**
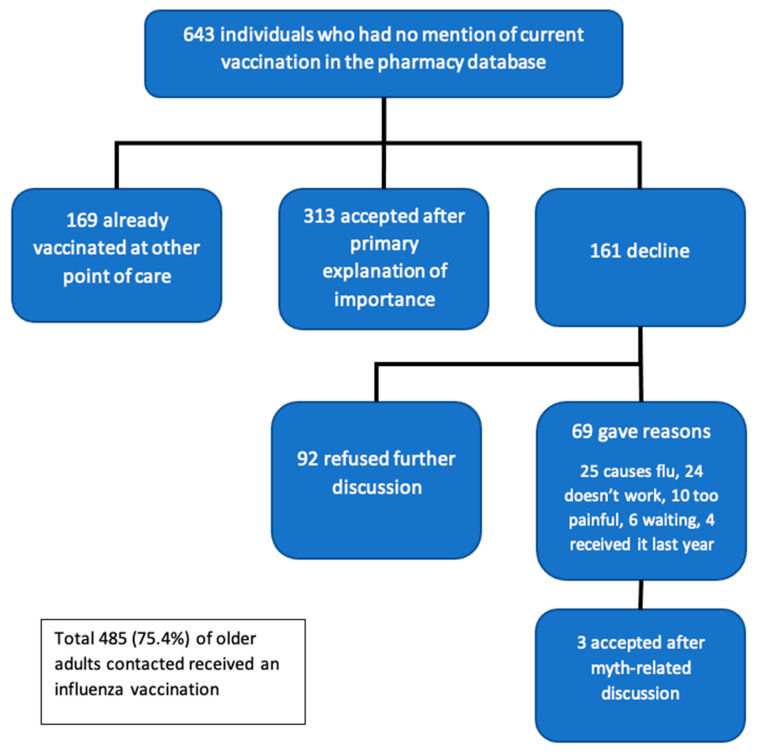
Patient pathway in the quality improvement project.

**Table 1 vaccines-09-00312-t001:** Reason for not wanting to receive the influenza vaccine.

Reason	Number of Older Adults (% of Patients Providing Reason)
“Flu shot causes the flu”	25 (36%)
“Flu shot doesn’t work”	24 (35%)
“Flu shot is too painful”	10 (14%)
“I would rather wait until the flu comes into the community”	6 (9%)
“I had the flu shot last year; I don’t need it again”	4 (6%)

## Data Availability

The data presented in this study are available on request from the corresponding author.

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
