# Peer review of "Achieving Influenza Vaccine Uptake Target in Canada via a Pharmacy-Led Telephone Discussion during the 2019–2020 Season"

_vaccines, 2021, doi:10.3390/vaccines9040312_

Round 1

Reviewer 1 Report

this project report addresses very important issue regarding vaccination against Influenza A. The project is impressive and the outcomes are encouraging.

I have only minor comments:

1- line 180: I think the word "table" at the start of the heading is not related to the sentence.

2- Lines 183-184: is this means that the pharmacy team could not address the reasons from rest of the participants? why and what are these reasons? Are there any actions regarding this; e.g. further education of the pharmacy team?

3- Are there any promotions given to the pharmacists to convince them participate and/or to continue in/with this project?

Author Response

Thank you for taking your time to review our manuscript and for your feedback. We have gone through these in detail and have made the appropriate changes to the manuscript. We have also modified the format of the manuscript to be in keeping with the template document we received. We, too, believe this project represents an additional tool in reducing vaccine hesitancy.

I have only minor comments:

1- line 180: I think the word "table" at the start of the heading is not related to the sentence.

This has been corrected.

2- Lines 183-184: is this means that the pharmacy team could not address the reasons from rest of the participants? why and what are these reasons? Are there any actions regarding this; e.g. further education of the pharmacy team?

Thank you for this observation, the reasons for refusal in those that were happy to discuss were recorded. We have explored these reasons with our focus groups and were planning to update the education pack for the 20/21 flu season. Clearly in the wake of the pandemic, flu vaccination has not been an issue, however, the relevant learnings and outcomes of the focus groups are currently being applied to reduce vaccine hesitancy in the COVID vaccination program and will be built into our 21/22 flu vaccine strategy. This will be the subject of a separate manuscript.

3- Are there any promotions given to the pharmacists to convince them participate and/or to continue in/with this project?

There were no promotions given to the pharmacists beyond the usual government reimbursement given to all pharmacists to administer the flu vaccine. We have included a comment to this effect in the methods.

“The community pharmacists did not receive any incentive beyond the usual government reimbursement for the administration of the flu vaccines for participation in this program.”

Reviewer 2 Report

It is a simple and straightforward text that reads through well. My single comment corresponds to the result in terms of reaching the 75% of coverage indicated by  WHO thanks to the phone intervention. However, according to the results, 169 people already received the vaccine from another provider and apparently the intervention did not influence their decision. Hence, those who decline it (161) plus those who receive it (313) should constitute the final target of the program and 313/474 = 66% of success rate of the program. I would not consider that the 75% threshold is reached by the intervention as this is only achieved when adding up the 169 plus the 313 and it is not correct, if I am not missunderstanding the calculations of the article.

Therefore, I suggest to re-write that part and differentiate the patients vaccinated thanks to the intervention and those that voluntarily got vaccinated in a different point of care. 

Author Response

Thank you for taking your time to review our manuscript and for your feedback. We have gone through these in detail and have made the appropriate changes to the manuscript.

We agree on re-reading that this was confusing. We have re-written this to be absolutely clear that the WHO 75% target, in those that had not come for their initial appointment or responded to initial invitation, was achieved by convincing two thirds of those that had not sought vaccination elsewhere to attend for an appointment. It is important to bear in mind that this was only in those who had not already received their vaccine.  Therefore, the 75% was achieved above and beyond the 100% who had already received their vaccination in the first wave.

It now reads

“In this population of 643 individuals who did not have a documented influenza vaccine from their primary provider, 169 (26.3%) had been vaccinated in another setting. Of the remaining 474, 316 (67%) agreed to receive the vaccine by the end of the telephone consultation (figure 2).  Thus, in total, 485 (75.4%) people received their influenza vaccine by the end of this intervention. As such, this quality improvement project demonstrated its primary target of achieving the WHO recommended 75% influenza vaccine uptake in older adults who had not been vaccinated by their usual care provider in the first wave.”